# Agent-DocEdit: Language-Instructed LLM Agent for Content-Rich Document Editing

**Te-Lin Wu[1]**,[*] **Rajiv Jain[2], Yufan Zhou[2], Puneet Mathur[2], Vlad I Morariu[2]**
[1]Character.AI, Menlo Park, CA, USA
`telinwu@character.ai`
[2]Adobe Research, San Jose, CA, USA
`{rajijain,yufzhou,puneetm,morariu}@adobe.com`

## Abstract

Editing content-rich and multimodal documents, such as posters, flyers, and slides, can be tedious if the edits are complex, repetitive, or require subtle skills and deep knowledge of the editing software. Motivated by recent advancements in both Large Language Model (LLM) agents and multimodal modeling, we propose a framework that automates document editing which takes as input a language edit request from the user and then performs sequential editing actions to the document the satisfy the request. Our proposed method, Agent-DocEdit, first grounds the edit request directly in the underlying document structure to identify the elements that need to be manipulated. Then, we rely on the agent capabilities of LLMs to generate an edit program which calls a set of pre-defined APIs to modify the underlying structure of the document. To improve the generated edit program, we leverage a feedback mechanism incorporating a deterministic code executor and a multimodal LLM. We demonstrate the effectiveness of our proposed modularized LLM editing agent on the DocEdit dataset, where Agent-DocEdit outperforms existing state-of-the-art methods by 70+% in document element grounding and 16+% on final rendition generation.

## 1 Introduction

Editing rich multi-modal content as found in flyers, graphic designs, posters, etc., requires knowledge of specialized tools and patience to apply multiple edits to image and text layers. Recent AI trends have increased interest in automating the process of editing such documents to reduce the skills and time required–powerful multimodal models are not only able to understand a user's language description of their edit intent but also generate new text (Achiam et al., 2023), visual content (Geng et al., 2023), or even code for carrying out the edits (Mathur et al., 2023).

There are two major approaches for applying recent techniques to document editing. The first is to model editing commands as structured and composable set of document elements, actions, and states and then analyze the document and user's description of their editing intent to produce an editing command. DocEdit (Mathur et al., 2023) is a recent example of this approach, and it uses a multimodal language model to understand the author edit request, identify the bounding box of the document element that is the target of the desired edit, and generate a short structured command. However, DocEdit does not model the underlying structure of the document (*i.e.*, the individual text, image, and shape elements along with the required information for composing them into a document) operating instead on the raster image of the document as input, nor does it carry out the edits to produce the edited document (*i.e.*, the command and bounding box are the final outputs). The second type of approach, such as InstructDiffusion (Geng et al., 2023), focuses on directly producing the resulting raster image of the edited document, resulting in an architecture that can be

---

[*] Work done during an internship at Adobe Research, Document Intelligence Lab (DIL).

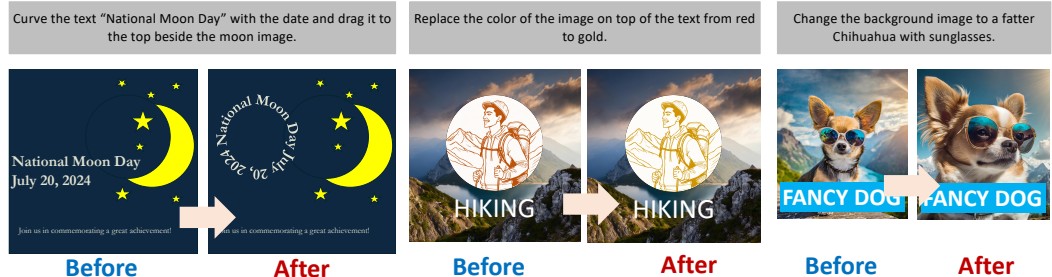

Figure 1: **Instruction based document editing:** Given a language user edit request as input (in the grey box), our framework automatically edits the *before* document to produce the *after* document. The illustrative examples above are similar to the training and evaluation data found in the DocEdit dataset (Mathur et al., 2023).

optimized end-to-end but which can generate artifacts or distortions due to the constraint that it can only manipulate the final pixels and must infer all the semantic groupings, constraints, and interactions among document elements as it performs modifications.

Building on recent LLMs based approaches such as Toolformer (Schick et al., 2024), we propose a method, `Agent-DocEdit`, that (1) takes as input a document with its full structure rather than just a raster image, (2) identifies within the document structure the elements involved in the edits (through a process called *grounding*), (3) produces an edit program to carry out the edits, and (4) executes it to produce a modified document structure that can be rendered by the document viewer if a raster of the document is desired, thus addressing the main weaknesses of the aforementioned approaches. One advantage of such an approach is that the edit program can leverage already existing and well-tested pre-defined APIs for editing documents rather than having to learn these transformations from scratch.

We formulate our problem as follows. Given a user editing request (in the form of natural language) and a document with multimodal content (a layout of text, shape, and image elements), the goal of our system is to automate the requested edits and produce the desired *edited document* by identifying in the underlying document structure the document elements involved in edits, producing an edit program to modify these document elements, and then executing the modifications through an API to produce a modified document structure that can be rendered to produce an image of the edited document. Figure 1 provides a few example editing commands as well as examples of before and after documents that our system is designed to work with. The first (leftmost) example will be used to illustrate different components of our proposed framework in the remaining sections. The proposed `Agent-DocEdit` framework is extensively evaluated on the *design subset* of the DocEdit dataset. Our main contributions are:

- We propose an LLM-based framework for document editing that, unlike previous work, operates directly on the structured representation of a document and produces structured content as output. This is done by grounding edit requests in the documents structure, generating an edit program, and executing the program through an external API to effectuate the desired edits. Compared to instruction-tuned diffusion baselines, we improve the image similarity-based metric by >16%.

- We propose two new techniques for document element grounding given a language instruction, both of which are consistent with our approach of using tools (where the visual phrase grounder is the tool). Both approaches significantly improve the ability of the system to localize the document elements involved in edits by >70% against the prior DocEdit baseline. This is particularly impactful, since an error in grounding results in the wrong document element to be modified.

- We propose a feedback mechanism that can improve the generated code by providing language feedback to guide the LLM to iteratively improve the generated plan, improving results by >2% in image similarity-based metric; and >35% in human judgements, against that without the feedback.

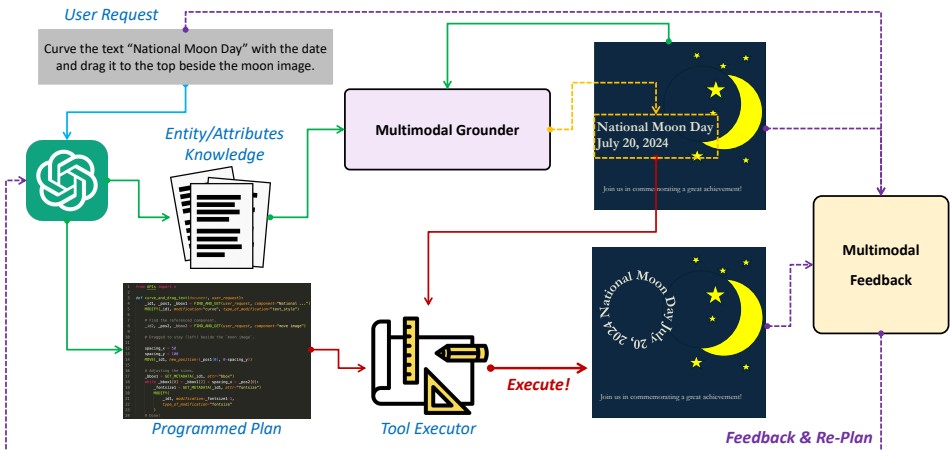

Figure 2: `Agent-DocEdit` is composed of: (1) an **Edit Program Generator** that transforms the user edit request to an edit program using an LLM, (2) a **Multimodal Grounder** that localizes the document elements referenced in the edit request, (3) an **Executor** that executes the edit program using pre-defined APIs which edits the content structure, and (4) a **Feedback Provider** that helps refine the edited outcome for improved results.

## 2 Method (The `Agent-DocEdit`)

Figure 2 provides an overview of the proposed `Agent-DocEdit` framework, which is composed of four main components: (1) the **Edit Program Generator** takes input human user editing request (in natural language) as input and generates the *edit program* (*i.e.*, a series of *pre-defined API* function calls as a high-level planning of the editing actions) by instructing an LLM to perform the code-generation task. (2) the **Multimodal Grounder** identifies the document elements referred to by the editing request, using an architecture derived from phrase grounding models for computer vision tasks. The ultimate output of the grounder will be the ID of the document element(s) within the document's content structure. (3) the **Executor** performs a deterministic conversion transforming the edit program to actual executable programs that modify with the content structure according to the editing request. (4) the **Feedback Provider** is a separately learned module that compares the edited outcome against the original edit request and produces language feedback that is then utilized to (iteratively) instruct the LLM to refine the generated edit program. We will now discuss each component in details below.

### 2.1 Generating an Edit Program via LLM-Code-Gen.

We instruct the LLM to produce a Python **edit program** which carries out the user edit request by calling a set of pre-defined APIs, acting as an *agent* that manipulates a set of provided *tools*. The API is provided to the LLM through the docstring of each function following similar designs in Surís et al. (2023) and Gupta & Kembhavi (2023). The docstring includes a description of the goal of the API function, input and output arguments, and exemplar usage. In our early studies, we empirically found that usage examples play a crucial role as *in-context examples* in instructing the LLM to produce higher quality (more accurate) programs. The detailed implementation of each API is not provided to the LLM, as the LLM can generate programs simply by following the API guidelines without needing to refer to how each function is implemented, *i.e.*, for the LLM to generate the edit program, it *only* needs to refer to the **API function header** as well as its **docstring context**.

#### 2.1.1 Pre-Defined APIs

In this work, we generally follow the *actions* defined by DocEdit (Mathur et al., 2023) (*i.e.*, MOVE, DELETE, ADD, COPY, MODIFY, REPLACE), with additional *localization* actions. We categorize these pre-defined API functions into three major categories:

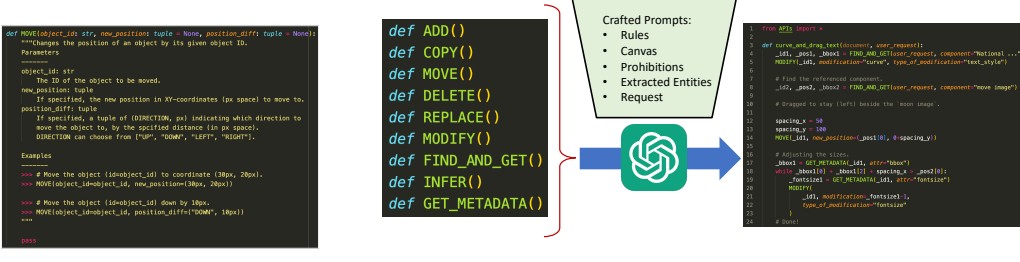

(a) Sample API  (b) Code-Gen.

Figure 3: **Edit Program Generation: (a)** A sample defined API function that performs the MOVE functionality. **(b)** An LLM is instructed to generate the edit program with the pre-defined APIs.

- **Localization Operations:** The **FIND** function performs entity extraction on the language request to identify which portions of the request refer to target document elements, and then it performs visual phrase grounding to localize the elements in the document. The underlying grounding model predicts bounding boxes, which are used to search over the document structure. The canonical ID of the document element with the highest intersection over union (IoU) will be returned.[1] In addition to the FIND function, we also implement an **INFER** function that predicts the destination location of a document element after editing for actions such as MOVE.

- **Spatial Operations:** Functions such as **MOVE**, **DELETE**, **ADD**, and **COPY** generally deal with spatial arrangements of the target components. Each function has its specific set of arguments, for instance, MOVE can take specified position delta or the actual target position to re-position a component. ADD takes object type (or description) to insert contents (either retrieved from an asset library or generated by a visual generative model) to a specified position, while DELETE simply takes the canonical object ID for deletion action.

- **Modification Operations:** Functions **REPLACE** and **MODIFY** perform content modifications (or swapping). The MODIFY function takes as arguments the type of modification (*e.g.*, text color or font-size) and the actual modification context (*e.g.red* for changing color) that will be acted on the target document element. REPLACE function will change a specified content (*e.g.*, a piece of text, an imagery) to a targeted type of content following the specification.[2]

By including these pre-defined APIs in the system prompt of an LLM, we instruct the LLM to perform *natural language to code* generation to derive a Python-based **edit program** where each action to be performed comes from one of the pre-defined APIs.

### 2.1.2 Edit Program Generation through Instructed Code-Gen.

We leverage the *instructed code-generation* capability of a strong LLM to generate a Python-based edit program. Given the user request in natural language, we instruct the LLM to follow a specifically crafted guideline for it to strictly refer to the pre-defined APIs described in Section 2.1.1 when devising the edit program. As illustrated, the left hand side of Figure 3 shows a sample API function (MOVE), where its right hand side exemplifies a sample generated edit program.

**Using Layout Canvas.** In this work, we find that defining the boundaries of the document contents (*i.e.*, the canvas where components are within) and supplementing the coarse level of content spatial locations, is helpful for the LLM to devise more accurate edit program. We thus include the canvas dimension of the document, locations (as bounding boxes) of major contents in the document (via PaddleOCR (Du et al., 2020)) in the LLM-instruction.

---

[1]The grounding model will *propose* several possible bounding boxes ranked with their logit scores, we simply take the top-1 scoring bounding box as the prediction.

[2]The REPLACE function can be viewed as a combination of DELETE and ADD, except that the added position is the same as the document element being modified.

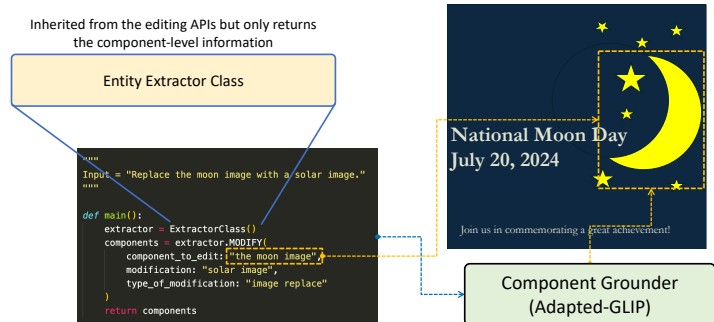

Figure 4: **Programmatic grounding:** Leveraging our pre-defined APIs, we construct a document element grounding pipeline to identify the components (document elements) of interest in the editing request, in order to facilitate the grounding in the actual documents.

## 2.2 Component Grounder

As described in Section 2.1.1, the FIND (and the INFER) function can be implemented via a visual phrase grounding model. In this work, we utilize GLIP (Li et al., 2022), an open vocabulary object detection model, to serve as the basis fulfilling such grounding requirement. However, most phrase grounding models require the *"phrase to ground"* to be specified beforehand, usually as a set of starting and ending spans of the phrases within the input sentence. We explore the two following methods for extracting the *phrase to ground* from the given user editing request:

**NLP Parsing.** The entity extraction task can be generally approached as a parsing task followed by simple heuristics. We use the semantic role labeling (SRL) (Shi & Lin, 2019) technique to parse the user request into a few arguments (ARGs) and then take the ARG1 as the main document element that needs to be edited. For instance, in the following SRL-parsed request: `"[V: Change] [ARG1: the date in second line] [ARG2: to 02.22.2024] ."`, we regard the ARG1 span (`the date in second line`) as the *"phrase to ground"*.

**Leveraging Pre-Defined APIs.** While parsing techniques such as SRL are specifically designed for textual entity extraction, we find that our code-generation framework also naturally supports such extraction capability. As illustrated by Figure 4, we devise a set of APIs that, when they are executed, the LLM is instructed to fill the function arguments with the extracted corresponding entities.

With the two aforementioned automatic entity extraction pipelines, we construct a fine-tuning dataset from DocEdit to adapt the GLIP model to our document editing domain, and utilize its predicted bounding boxes for the target component localization. Note that the entity extraction pipeline will be absorbed into part of the FIND and INFER functions.

## 2.3 Execution

Once the edit program is generated, each major **spatial** and **modification** API operations will call its corresponding executable implementations to perform the actual editing process. These operations are deterministic and are editing software-dependent, while the major API operations defined in Section 2.1.1 are generally applicable and hence software-agnostic.[3]

The execution of FIND and INFER are similar, the predicted bounding boxes of the target component will be used to search over the document structure and return the canonical component ID with the highest IoU score. Such IDs will be the major medium used for other APIs to refer to a specific component. For ADD and REPLACE, there are some occasions that new content needs to be generated (*e.g.*, *changing the background image to something else*). We

---

[3]The main editing software used in this work implements a TypeScript-based low-level API library that can fulfill all the functionalities our pre-defined APIs require, where these low-level APIs will eventually be able to interact with the underlying document structure and perform the actual component manipulations/editing.

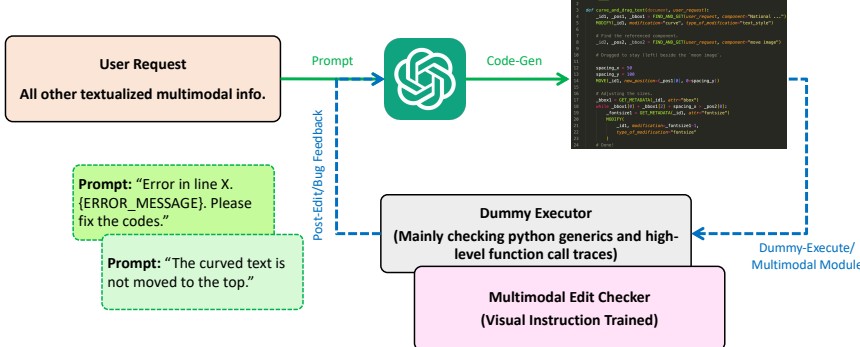

Figure 5: **The feedback mechanism** mainly concerns two types of errors: (1) **syntax error**, which can be resolved by re-prompting the code-gen. LLM with extracted standard program execution error (STDERR), (2) **semantic error**, which is resolved by leveraging a trained *multimodal feedback provider*, for guiding the re-generation of the edit program.

execute such request by feeding the extracted component (*i.e.*, the extracted entity should be the content to be added or replaced with) to a text-to-image generative model as its prompt.

## 2.4   Feedback Mechanism

The LLM-code-generation pipeline in Section 2.1.2 often generates high-quality edit programs to fulfill user requests, but these programs can occasionally produce errors (bugs) that either prevent the program form executing (syntax errors) or which produces unsatisfying results (semantic errors).

- **Syntax Error:** Although as mentioned in Section 2.1.2, we instruct the LLM to perform code-generation strictly following the defined APIs and their argument structures, we find that the LLM can occasionally *hallucinate* how these functions are used. Concretely, we find that the LLM can hallucinate unspecified usages of the functions (*e.g.*, using arguments that are not specified) or even unspecified API calls (*e.g.*, the LLM can have a tendency to *invent* nonexistent APIs).

- **Semantic Error:** This type of error occurs when the edit program is syntax bug-free, but exhibit behaviours that are not perfectly satisfying the user editing request. For instance, a request of moving a target component upwards but below another reference component, might result in a generated program that accidentally moves the component too far to the top. Another representative (and major) error is when the component grounder fails to localize the target component perfectly. In such cases, even though the generated edit program is semantically correct, the outcome can still be suboptimal because the program is applied to the wrong document element.

We address these issues with a feedback mechanism, as illustrated in Figure 5. For a syntax error, we devise a dummy executor that executes the generated edit program without calling the executable API, *i.e.*it calls a dummy API for the purpose of checking the syntax correctness of the generated edit program. Whenever an error occurs, we extract the error message (STDERR) and feed it back to the LLM to obtain the corrected edit program via a *re-prompting* (see Figure 5). We bound the number of re-prompting trials maximally at 5.

Since the second type of error, the semantic error, will only be recognizable after the editing outcome is produced, and furthermore, there is no real deterministic *"compiler"* to identify whether the outcome is satisfactory or not, we propose to learn a multimodal model for providing semantic error feedback. Specifically, we train a multimodal LLM (Figure 6) to predict the type of semantic error (*i.e.*, whether the error comes from semantic mistake of the edit program or errors from the grounder) by analyzing the edit request along with the before- and after-edit document renditions.

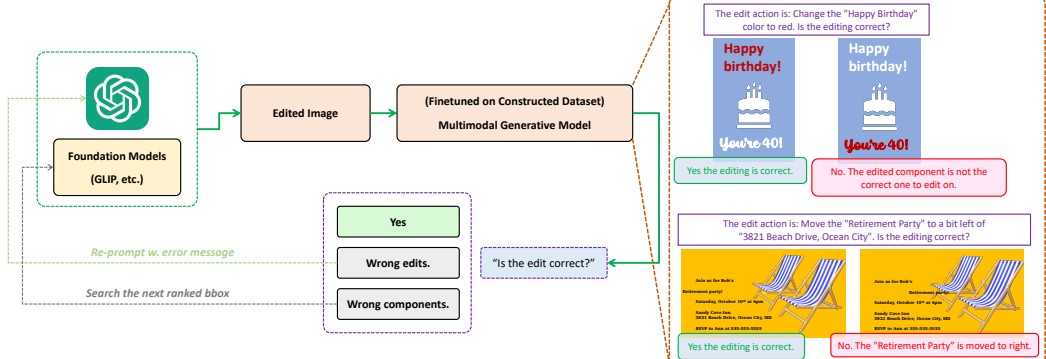

Figure 6: **Multimodal feedback provider** is trained with synthetically constructed datasets as shown on the right hand side. We utilize our programmable pipeline to construct positive and negative pairs of edited outcomes associated with the original request, in order to train a multimodal LLM to provide the feedback (in language).

### 2.4.1 Learning The Multimodal Feedback Module

To learn the multimodal feedback module, we use a popular open-source LLaVa model (Liu et al., 2024) and extend it to take paired images (*i.e.*, before- and after-edit document content) to provide natural language feedback of the editing outcome. To equip the model with capability of inferring whether an edited outcome is satisfactory, we train it with correct (positive) and incorrect (negative) results of the editing actions. Since each API defined in Section 2.1.1 has a corresponding executable implementation, we can construct a deterministic mapping between a series of these API calls with the actual outcome after the execution of such API series.

**Dataset Construction.** We craft a set of *templated* editing requests such as "[MOVE] {component$_a$} to [direction] of {component$_b$}", or "[REPLACE] {component$_a$}'s {color} to [red]". Given document content, items within the braces ({}) will be replaced randomly with the document element to be manipulated and its corresponding attribute(s) to edit. Items in the brackets ([]) are sampled from a pre-defined set of allowed vocabulary, *e.g.*, REPLACE can have synonyms such as CHANGE, and colors can be sampled from red, blue, green, and more. It is noticeable that a deterministic mapping can be constructed between the editing APIs and these templated requests. As illustrated in the right hand side of Figure 6, we construct a dataset based on the templated editing requests, where a negative outcome is generated by by deliberately causing the API to incorrectly apply the desired edit request (*e.g.*, sampling the wrong [direction], [color] or the incorrect {component}). Therefore, the negative samples can be constructed by manipulating the contents with exactly opposite or wrong actions (*i.e.*, the right/wrong actions are known beforehand from the ground truths) against a sampled request – guaranteeing the resulting samples contradict it.

The fine-tuned LLaVa model is trained on producing a simple *yes/no*-based feedback, which can also be deterministically written based on constructed positive or negative pairs.

**Feedback-based refinement.** Once the natural language feedback is obtained following the procedure in Section 2.4.1, we can provide such feedback to the LLM to refine the generated edit program. We only care about negative feedback, which is of two main types: (1) For **semantically wrong editing** we perform a "re-prompting" of the LLM to generate the refined program with the instruction depicted in Figure 5. (2) For **wrongly grounded component**, we sample the next highest ranked proposed bounding box from the grounder (Section 2.2) without modifying the LLM generated program. The underlying assumption is: at least one of the proposed boxes would correctly localize the target component, which is not necessarily true in all the cases. We leave the general cases as future work.

# 3 Experiments

Our experiments seek to answer following research questions: (1) How does the proposed modularized editing-agent framework compare to the baseline end-to-end model reported in Mathur et al. (2023) on its various evaluation metrics? (2) How does the proposed framework compare to other instruction-tuned generative editing models? (3) How effective is the proposed feedback mechanism?

**Implementation Details.** For the document element grounder, the GLIP model (Li et al., 2022) is adopted and extended from the original author's repository and pretrained model weights. For the code-generator, as well as the entity extractor, we use OpenAI's GPT-4 model (Achiam et al., 2023). For the image generative model, we use the stable diffusion version 1.5 (SD-v1.5) (Rombach et al., 2022) and cap the denoising steps at 50. We adopt the LLaVa version 1.5 (Liu et al., 2024) with the language model part being LLaMa-2-13b (Touvron et al., 2023) for the feedback mechanism in Section 2.4.1.

## 3.1 Experimental Setups

**DocEdit Metrics.** For various component-wise and user request intent understanding metrics, we use the ones reported in the original DocEdit work (Mathur et al., 2023). Specifically, the two main evaluations are conducted on the **accuracy** of both action parameters (**Action Para.**) and component parameters (**Component Para.**), where the parameters here refer to the type(s) of actions/components the user editing request concerns. As mentioned in Section 2.1.1, the design of our pre-defined APIs follow the major *actions* defined in the DocEdit dataset, and hence we regard our generated edit program as correct on predicting the action parameters if a line of the generated program contains the API names same as the ground truth action. Similarly for the *components*, we evaluate the extracted component types from the grounder (Section 2.2) against the ground truth.

**Image Similarity Metrics.** In addition to the original DocEdit defined metrics, we further evaluate how similar the edited outcome from our framework is, compared to the ground truth post-edit documents. Motivated by (Geng et al., 2023), we extend the CLIP-Score (Hessel et al., 2021) to consider the relationship between the edit request and the before/after-edit changes to the document (as an image). Specifically, we use the following formula to quantify the similarity measures ($\phi$ indicates cosine similarity from either *CLIP* visual embeddings or text-to-image *CLIPScore*):

$$\phi^{CLIP}(\mathbf{e}_{\text{img}_{gt}}, \mathbf{e}_{\text{img}_{edit}}) - \|\phi^{CLIPScore}(\mathbf{e}_{\text{img}_{gt}}, \mathbf{e}_{\text{text}}) - \phi^{CLIPScore}(\mathbf{e}_{\text{img}_{edit}}, \mathbf{e}_{\text{text}})\|_1$$

In addition to using the entire document image for comparisons, we also report the similarity measures focused only on the contents within the ground truth bounding box(es) (**Boxed-**).

**Instruction-Tuned Editing Baselines.** Our two instruction-based editing baselines are: InstructPix2Pix (Brooks et al., 2023), and InstructDiffusion (Geng et al., 2023). They are similar in that they both accept a human language instruction on how the images are to be edited, while a text-to-image diffusion model (Rombach et al., 2022) is trained to follow such an instruction to *generate* an **edited version** of the input image. While InstructPix2Pix mainly leverages the Prompt2Prompt technique (Hertz et al., 2022) for the editing data curation, InstructDiffusion adopts a wider range of more sophisticated data construction methods for editing tasks such as *object removal/replacement*, *image enhancement*, and *style transfer*, making it an improved version of InstructPix2Pix.

## 3.2 Results

**Quantitative Results.** Table 1 summarizes the essential DocEdit performance metrics. It can be observed that for the **Action Para.** metric, our framework (AD as `Agent-DocEdit`) performs somewhat worse (66.7) than the original DocEdit model. However, an empirical inspection suggests that the generated edit program can actually be correct even if the main APIs used do not exactly match the ground truth action parameter (*e.g.*, `REPLACE` can be a combination of `DELETE` and `ADD`, and `MODIFY` and `REPLACE` can be equivalent for *textual* editing).

| Models | Action Para. | Component Para. | Code 1-Pass | Code-5-Pass (w. Feedback) | Sem. Err. |
|---|---|---|---|---|---|
| DocEdit | 84.5 | 52.5 | — | — | — |
| `Agent-DocEdit` | 66.7 / **96.7** | 77.8 | 66.7 | — | 34.6 |
| `AD w. Feedback` | | **87.4** | 66.7 | **95.3** | **12.3** |

Table 1: **Components Performance:** the main evaluation metrics in the Doc-Edit task against its end-to-end trained baseline. *Slash under **Action Para.** indicates the re-definition of the ground truth action parameters.

| Models | AP | AP50 | AP75 | Top-1 Accuracy |
|---|---|---|---|---|
| DocEdit | — | — | — | 34.34 |
| NLP (SRL) | 27.16 | 54.82 | 22.60 | 58.37 |
| Code-Gen. | **32.57** | **59.19** | **30.60** | **59.85** |

Table 2: **Component grounding performance.**

| Models | Directional↑ | Boxed-Directional↑ |
|---|---|---|
| InstructPix2Pix | 0.741 | 0.781 |
| InstructDiffusion | 0.806 | 0.792 |
| `Agent-DocEdit` | 0.921 | 0.886 |
| `AD w. Feedback` | **0.939** | **0.896** |

Table 3: **Editing performance** (similarity-based).

We thus establish a mapping (Append. Sec. A.1) between equivalent DocEdit action parameters and the results in Table 1 (after the slash) indicates that our LLM pipeline is actually significantly better than the baseline end-to-end trained model. Our framework outperforms the baseline by a large margin in the component parameter prediction, where the feedback mechanism improves further.

| Models | Avg. Rank↓ |
|---|---|
| InstructPix2Pix | 3.43 |
| InstructDiffusion | 3.18 |
| `Agent-DocEdit` | 1.89 |
| `AD w. Feedback` | **1.14** |

*Rank is among 1-4.

Table 4: **Human preference** (mean ranked).

The edit program generator averagely generates 10.75 lines of code with 1.4 major API calls. The semantic error rate (number of errors/edits) within a held-out human judged subset is 34.6% without feedback and 12.3% with feedback, where the feedback provider reaches 71% accuracy on a held-out test set. In Table 1 we also report the passing rate (bug-free during execution) of the generated edit program, the syntax error re-prompting within 5 trials can improve the passing rate to almost perfect. These results suggest that our feedback mechanism significantly reduces both syntax and semantic errors in the edit programs.

In addition to the results reported in Table 1, we also examine the individual semantic error rates per each major API action. The error rates (without feedback mechanism) of the top three frequent actions in a held-out subset are: REPLACE: 22%, MODIFY: 30%, and MOVE: 37%. It is noticeable that our system does particularly well at the action REPLACE, and we hypothesize MOVE can benefit from improved target position predictors. We also observe that our component grounder performs quite evenly well on all kinds of components – no particular bias against or towards certain kinds of components is observed.

The comparisons between the two grounding methods in Section 2.2 is reported in Table 2, where a significant improvement can be observed using the code-generation method with our defined APIs. Both methods perform well above the end-to-end baseline.

Table 3 reports the performance for the similarity-based editing scores. For fair comparison, if the generated edit program fails during the execution (with syntax errors), we do not proceed with any editing to the document, *i.e.*, a failed program makes the content *unchanged*. It is notable that our framework, with or without the feedback mechanism, surpasses the two instruction-tuned baselines. The results reveal our general observations, which are: The two (instruction-tuned) generative baselines often exhibit severe degeneration of the parts that are supposed to be intact, or fail to capture the intentions of the user editing request. Our modularized agent framework more successfully interprets the user requests and manipulates the components much more accurately. And moreover, the feedback mechanism alleviates initial errors from both the edit program generation as well as inaccurate grounding.

To verify these numerical results align with human judgements, we further conduct a human study on a randomly sampled 100 subset, each ranked by three annotators of which model outcome is the more preferred, and averaged as in Table 4. The human preference trends verify the aforementioned results. Note that human judgement prefers the outcome with

our feedback mechanism significantly more than that without (the gap between them is also larger than that of the similarity-based metric). This is reasonable since similarity-based metric may not be very detail-oriented in judging our task.

## 4 Related Work

**LLM Agent.** Equipping an LLM with the ability to use external tools makes it a powerful model that enjoys both strong generalization of language modeling capability as well as the faithful information manipulation. Prior work such as ToolFormer (Schick et al., 2024), TALM (Parisi et al., 2022), and program-aided language models (Madaan et al., 2022; Chen et al., 2022; Gao et al., 2023; Wang et al., 2022c), successfully learn an LM that is able to manipulate *tools* such as calculators, calendars, and programs, for achieving desired tasks. Program-based LLMs are also applied to control tasks, such as ProgPrompt (Singh et al., 2023), and Eureka (Ma et al., 2023). One type of tool-equipped LLM is able to perform *visual programming*, *i.e.*, generate programs that call APIs, such as object detectors, to deal with visually-related queries, such as VQA (Gupta & Kembhavi, 2023). Our work is most closely related to ViperGPT (Surís et al., 2023), where we extend the visual programming ability to content editing with additionally created feedback mechanism.

**Document/Content Understanding.** Our work is also related to document content understanding, as well as layout comprehension. Prior works have investigated applying modern transformer architecture for an end-to-end trainable document processing models, such as DocFormer (Appalaraju et al., 2021), text-image-layout (Powalski et al., 2021), LayoutLMv2/3 (Xu et al., 2021; Huang et al., 2022), Dessurt (Davis et al., 2022), and UI/layout parsing or understanding (Lee et al., 2023; Li & Li, 2023; Wang et al., 2022b) Along this line of work as well as the grounding capability (Deng et al., 2021; 2023), the DocEdit dataset also features an end-to-end learnable model (Mathur et al., 2023) (in this work we use GLIP (Li et al., 2022)). We show that modularized framework leveraging LLM as an agent to *program* the edit actions with the help of external tools (*e.g.*, component grounders) is a strong competitor for understanding and editing content-rich documents.

**Content Editing.** Our work is closely related to learnable content editing models. Several prior works have also studied language-instructed (Chen et al., 2018; El-Nouby et al., 2018; Li et al., 2020; Lin et al., 2020b; Jiang et al., 2021a) or conversational (Lin et al., 2020a; Jiang et al., 2021b) image-editing, where a language processing module is learned to estimate the requested intents to guide the image-editing tasks. Recent works have adopted modern transformer architecture for better facilitate the instruction comprehension and planning (Shi et al., 2021), as well as diffusion models for the edited outcome generation (Wang et al., 2022a; Hertz et al., 2022). Notable works, including InstructPix2Pix (Brooks et al., 2023), InstructDiffusion (Geng et al., 2023), and Emu-Edit (Sheynin et al., 2023) push further the generative editing performance with large-scale and multi-tasking data. Inspired by the prior literature, as well as the motivation of the tool (grounding and editing software) usage and keeping unedited parts intact, our framework is able to more seamlessly (and faithfuly) process wider-range of editing actions.

## 5 Conclusion

In this work, we propose a multimodal LLM-agent-based framework, `Agent-DocEdit`, to tackle the task of automatic (multimodal) document content editing. The proposed framework is composed of four main components: (1) The **Edit Program Generator** that translates the user editing request to an edit program that calls a set of pre-defined APIs for basic content manipulation actions; (2) The **Multimodal Grounder** extracts the key entities within the request and localize their corresponding document elements within the document structure; (3) The **Executor** executes the edit program by calling a set of pre-defined APIs to interact and manipulate the document structure to obtain the actual edited outcome; (4) The **Feedback Provider** takes as inputs the request and before-/after-edit documents to provide potential refinement feedback for generating more satisfying outcome; We evaluate `Agent-DocEdit` on an editing dataset, DocEdit, and demonstrate its effectiveness over the existing DocEdit baselines and strong instruction-tuned diffusion-based editing models.

## Ethics and Broader Impacts

We acknowledge that all of the co-authors of this work are aware of the provided *ACM Code of Ethics* and honor the code of conduct. This work is about constructing a modularized multimodal framework (`Agent-DocEdit`) that can comprehend human user's editing request to a given document content, where the framework automates the editing process by using LLM's code-generation ability as well as external tools to fulfill the request. Below, we consider ethical considerations and our potential impacts to the community.

**Dataset.** The dataset and the document content used in this work are obtained from a published prior work, DocEdit, where human annotators helped creating an editing request dataset paired with corresponding outcomes. We do not foresee any harmful effects that can be potentially caused to the users as the dataset merely contains poster and flyer-like documents for common activities such as advertising, celebrations, announcements, and similar situations.

**Techniques.** The technique developed in this work is generally benign to the users, unless the user indicate any harmful request (*e.g.*, generating disturbing contents and/or re-arranging the textual contents to harm recipients). Our framework is learned and trained mainly using the DocEdit dataset, which possess no intended harms from the content creators.

**Limitations.** In this work, our edit program APIs are designed to be generally applicable and software-agnostic. However, we also mentioned that the detailed contents of these APIs can have software dependencies. That is, those APIs are not exactly the final interfaces for manipulating the underneath document structure/contents, where some efforts would be made to map them to the actual execution functions (*e.g.*, mapping from our Python APIs to corresponding design editing software code.)

Although our modularized framework can process multiple contents by compiling a procedure invoking multiple API calls (with proper logic) to the contents, the user requests, for this work, may need to be more specific. For example, for implicit command such as *"Change the daytime scene to night time."*, the user may need to elaborate the meaning of "daytime" scene (*e.g.*, to something more explicit like *"Change the moon to the sun."*) as inferring implicit editing is left as a future exciting work that we are intrigued to explore as well as hope to inspire the community to work on.

We also would like to acknowledge a limitation where this work implicitly assumes the structure of the so-called content-rich documents, with an emphasis on presumption of existing editing software. We foresee a combination of pixel-generative models and our framework as an exciting future direction for achieving even more general use cases.

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

## A  More Details on Utilizing DocEdit Dataset

### A.1  Action Parameter Mapping

As mentioned in Section 3.2, under certain circumstances, some API functions can be equivalent to one another of a combination of a few other APIs. Below we attach such mapping for the ground truth action parameter re-definition, for more fairly evaluate our LLM-agent framework.

```
action_mappings = {
    "REPLACE": [
        "DELETE->ADD", "ADD->DELETE"
    ]
}

component_based_action_mappings = {
    "TEXT": {
        "REPLACE": ["MODIFY"],
        "MODIFY": ["REPLACE"],
    }
}
```

The mappings above indicate that the replacement REPLACE can be a combination of deletion (DELETE) and addition (ADD), where the order does not matter. We also include a type of mapping that is component dependent, as for textual components (TEXT), the replacement and modification are essentially equivalent.

### A.2  Data Splits

For both the grounding and feedback mechanism training, we use only the training set from the *design subset* of DocEdit, where the evaluations are done on its original testing set.

### A.3  Human Studies

We recruited five college students with linguistic/CS backgrounds and presented them with the edit request alongside outcomes from all four methods (scrambled). They ranked them (1-4) on how aligned the samples were to the request. Each instance is ranked by all the evaluators and averaged as in Table 4.

## B  More Details on the Proposed `Agent-DocEdit` Framework

### B.1  Additional Implementation Details

We use the AllenNLP SRL package (Gardner et al., 2017) for NLP parsing entity extraction methods mentioned in Section 2.2.

For the document element grounder, the GLIP model (Li et al., 2022) features the language modeling part with a BERT-large (Devlin et al., 2019) model, and a vision encoder with Swin-L vision transformer model (Liu et al., 2021)).

In our preliminary studies, we experiment finetuning the feedback mechanism LLaVa model with the LoRA (Hu et al., 2022) variant and the whole finetuning variant, where we do not find significant performance difference in our task. Consequently, to reduce training cost, we only use the LoRA version thereafter.

### B.2  Instructions for Edit Program Generation

We hereby provide all the APIs used in Section 2.1.1 below:

```python
def MOVE(object_id: str, new_position: tuple = None, position_diff: tuple
    = None):
    """Changes the position of an object by its given object ID.
    Parameters
    -------
    object_id: str
        The ID of the object to be moved.
    new_position: tuple
        If specified, the new position in XY-coordinates (px space) to
    move to.
    position_diff: tuple
        If specified, a tuple of (DIRECTION, px) indicating which
    direction to
        move the object to, by the spcified distance (in px space).
        DIRECTION can choose from ["UP", "DOWN", "LEFT", "RIGHT"].

    Examples
    -------
    >>> # Move the object (id=object_id) to coordinate (30px, 20px).
    >>> MOVE(object_id=object_id, new_position=(30px, 20px))

    >>> # Move the object (id=object_id) down by 10px.
    >>> MOVE(object_id=object_id, position_diff=("DOWN", 10px))
    """

    pass

def DELETE(object_id: str):
    """Deletes or removes an object by its given object ID.
    Parameters
    -------
    object_id: str
        The ID of the object to be deleted.

    Examples
    -------
    >>> # Delete the object (id=object_id).
    >>> DELETE(object_id=object_id)
    """

    pass

def ADD(object_type: str, object_description: str, position: tuple = None
    ):
    """Adds an object by the specified type.
    Parameters
    -------
    object_type: str
        The type of the object to be added.
    object_description: str
        The description of the object that is to be generated or
    retrieved.
    position: tuple
        If specified, the position in XY-coordinates (px space) to add
    the object.
        If not specified, add the object to a random position.

    Examples
    -------
    >>> # Add a bullet point object.
    >>> ADD(object_type="BULLET")
    """
```

```python
    pass

def COPY(object_id: str, position: tuple = None):
    """Duplicates an object by its given object ID.
    Parameters
    -------
    object_id: str
        The ID of the object to be deleted.
    position: tuple
        If specified, the position in XY-coordinates (px space) to add
        the newly duplicated object.
        If not specified, duplicate the object to a random position.

    Examples
    -------
    >>> # Copy a text box object.
    >>> COPY(object_id=object_id)
    """

    pass

def REPLACE(
    object_id_to_replace: str,
    type_of_object_replaced: str,
    new_object_type: str,
    new_object_content: str
):
    """Replaces an object by its object ID with a newly added object by
    its type and content description.
    Parameters
    -------
    object_id_to_replace: str
        The ID of the object to be replaced.
    type_of_object_replaced: str
        The type of the object to be replaced.
    new_object_type: str
        The type of the new object to be added to replace the original
    object.
    new_object_content: str
        The content or description of the new object to be added to
    replace the original object.

    Examples
    -------
    >>> # Replace a text box object with an image of a cat.
    >>> REPLACE(object_id_to_replace=object_id, new_object_type="image",
    new_object_content="a cat")
    """

    pass

def MODIFY(object_id_to_modify: str, modification: str,
    type_of_modification: str):
    """Modifies the content or attributes of an object.
    For text objects the modification is the textual content to modify or
     replace the original one.
    For non-textual objects the modification is the edited attribute to
    the object being modified.
    Parameters
    -------
    object_id_to_modify: str
        The ID of the object to be modified.
```

```
    modification: str
        The modification to be made.
    type_of_modification: str
        The type of modification to be made, for example, text, color,
    font-size, shape, and font-style.

    Examples
    -------
    >>> # Modify a text box object with the modification texts.
    >>> MODIFY(object_id_to_modify=object_id, modification="One year old
    .", type_of_modification="text")
    >>> # Replace the background color from green to blue.
    >>> MODIFY(object_id_to_modify=object_id, modification="blue",
    type_of_modification="color")
    """

    pass

def FIND_AND_GET(component_to_edit: str, document=None):
    """Finds the component objects with the given component names or
    descriptions
    and gets the component ids, positions, and bounding boxes.
    Parameters
    -------
    component_to_edit: str
        The description of the component to be edited.
    document:
        The object for the slide document.

    Examples
    -------
    >>> # Get the component id of "the dashed lines under the image."
    >>> # from the user request: "Remove the dashed lines under the image
    ."
    >>> line_id, line_position, line_bbox = FIND_AND_GET(
    >>>     component_to_edit="the dashed lines under the image",
    >>>     document=document,
    >>> )
    """

    return id, position, bbox

def INFER(component_to_edit: str, document=None):
    """Predicts the position and the bounding box of the given component
    name
    or descriptions.
    Parameters
    -------
    component_to_edit: str
        The description of the component to be edited or added.
    document:
        The object for the slide document.

    Examples
    -------
    >>> # Predict the final position of the moved "dashed lines"
    >>> # from the user request: "Move the dashed lines under the image."
    >>> postedit_line_position, postedit_line_bbox = INFER(
    >>>     component_to_edit="dashed lines",
    >>>     document=document,
    >>> )
    """
```

```python
    return postedit_position, postedit_bbox

def GET_METADATA(component_id: str, attr: str, document=None):
    """Returns the metadata information of the given component.
    Parameters
    -------
    component_id: str
        The canonical id of the component to get the metadata from.
    attr: str
        The type of the metadata attribute to return.
    document:
        The object for the slide document.

    Examples
    -------
    >>> # Returns the metadata information such as `fontsize`, or `bbox`.
    >>> bbox = GET_METADATA(
    >>>     component_id="some id",
    >>>     attr="bbox",
    >>>     document=document,
    >>> )
    """

    return information
```

And additionally, the actual instructions to the OpenAI GPT-4 code-generation pipeline is as follows:

```
```python
[INSERT APIs HERE]

document = Document("{path/to/the/document}")
```

Write a function using Python and the functions above that could be
    executed to perform edits on the provided document template.

Consider the following guidelines:
- Use base Python (comparison, sorting) for basic logical operations,
    left/right/up/down, math, etc.
- Write only a single function with the name "main".
- No need to deal with output saving.
- Do NOT use APIs outside of provided ones and generic Python calls.
- Do NOT execute the function.
- object\_position is anchored at top-left corner, object\_position[0] is
    x coordinate and object\_position[1] is y coordinate.
- object bbox is anchored at its top-left corner, bbox = [x coordinate, y
    coordinate, width, height].

Command: "[USER EDIT REQUEST]"

The document width x height is [HEIGHT] x [WIDTH]
[INSERT CANVAS INFO.]
```

### B.3 Hyperparameters

Table 5 summarizes the main hyperparameters in all of our learning modules.

| Models | Batch Size | Initial LR | # Training Epochs | Gradient Accumulation Steps |
|---|---|---|---|---|
| Grounder (GLIP) | 8 | $1 \times 10^{-3}$ | 5 | 1 |
| Feedback Provider (LLaVa-1.5) | 16 | $2 \times 10^{-4}$ | 3 | 1 |

Table 5: **Hyperparameters in this work:** *Initial LR* denotes the initial learning rate. All the models are trained with Adam optimizers (Kingma & Ba, 2015). We include number of learnable parameters of each model in the column of # *params*.

