# OpenReview forum: "Agent-DocEdit: Language-Instructed LLM Agent for Content-Rich Document Editing"
_colmweb.org/COLM/2024/Conference — COLM_

### Official Review · Reviewer_4fVM · 2024-05-09

**Rating:** 9
**Confidence:** 5
**Ethics Flag:** 1

**Summary:**

The authors report on a very interesting and very relevant problem related to document engineering: enabling users to perform complex document editing tasks which can be formulated as short natural language requests or edits. The use of LLMs to transform these requests into sequences of operations calling various already existing APIs is very convincing and the evaluation shows encouraging results. The paper is in general well written and easy to follow, the running example is well chosen.

What is not clear from the evaluation, however: is there any comparison on human execution of the requested edits? Would humans reacting to feedback to first drafts usually given in short notes or sentences just like used here, execute similar or other actions? Would the result be similar or identical? I think the emphasis would not be in efficiency as in "getting more edits done in less time so paying layouters less" but to be able to present more variants easily to a client to show different solutions to a more general feedback like "can you do something about X here".

I have two additional issues the authors should resolve for submitting the final paper, see below.

**Questions To Authors:**

Please position the section on related work just after the introduction. This way readers can grasp why you are using which existing components. The introduction with the very short reference to the two existing but unsatisfactory approaches is fine as it is now.


The formulation "linguistic request" is rather odd: One would assume edit requests related to linguistic features of text like shortening sentence or paragraphs, reversing arguments, changing from passive to active voice, etc. On some instances you use the phrasing "natural language" or "language"—which is more appropriate. In fact, the main advantage is that users can formulate ediing requests by using unrestricted natural language. Please change the respective instances in the text accordingly.

**Reasons To Accept:**

The use of LLMs to transform these requests into sequences of operations calling various already existing APIs is very convincing and the evaluation shows encouraging results.

**Reasons To Reject:**

There are no reasons to reject the paper

---

> ### Author Rebuttal · Authors · 2024-05-31
>
> We are thankful to Reviewer 4fVM for the very positive feedback, where our proposed framework is very convincing in the realm of content-rich document editing. We address the comments below.
>
>
> __[W0] Comparison to human executions.__
>
> We would like to point out that the ground truth actions/components from the DocEdit dataset reflect what the annotators believed would be the actual actions for the edit requests.
> As a result, the actions performed that our framework is trained to mimic is __"highly aligned"__ with the actual human execution.
>
> The DocEdit dataset itself also features the modified (edited outcome) documents, representing the results of the human executions.
>
> As for the format of the feedback, we make the sentences short and concise so our framework can perform more deterministic revisions, and since they are deterministic, we do believe that human editors would take similar actions upon receiving the same feedback. However, we also think it is an interesting extension of ours to cope with (and train) more complex feedback (provider), where some learnable modules will be integrated to comprehend the more unrestricted natural language feedback.
>
>
> __[Q1] Paper refinements and section rearrangements.__
>
> Thank you for the suggestion on restructuring our paper. We will follow it when revising the final manuscript.
>
>
> __[Q2] The phrase regarding “linguistics”.__
>
> The _”linguistic request”_ here refers to the _“textual (or verbal) request”_ as some editing requests could be non-verbal (such as dragging the contents directly). We apologize for the misuse of the particular terms and will revise them according to your suggestions.

---

> ### Comment · Reviewer_4fVM · 2024-06-06
>
> Thanks for the response!

---

### Official Review · Reviewer_gr6V · 2024-05-10

**Rating:** 6
**Confidence:** 3
**Ethics Flag:** 1

**Summary:**

This paper reads like an industry-track system description paper about a system architecture for document editing. There are 4 system components working together to allow users to edit visual documents (in this work: an image rendered from an underlying document structure). An interesting feature of the system is a learned feedback loop that yields pretty impressive gains over the baseline.

**Questions To Authors:**

- Not sure about the term "grounding" as this usually has a different meaning in the context of NLP/LLMs.

**Reasons To Accept:**

The "system description" flavor is not necessarily a bad thing. The proposed architecture seems thought through well and fit for real world challenges. The reported results showcase nice gains over reasonable baselines on existing published datasets.

**Reasons To Reject:**

Even though the overall evaluation metrics look good, I would have wished for more ablations. There are many moving parts to the system, and breaking it down would give valuable insights regarding all the design decisions that eventually led to the proposed architecture. Aspects like the complexity (or simply lines of code) of the generated edit programs, the number of syntax and semantic errors without feedback mechanism, the accuracy of the semantic error classifier, etc. would give the reader a better intuition.

My two main concerns, however, are about clarity and limitations. Sec. 2.3 states that the API operations in the edit program "are deterministic and are editing software-dependent, while the major API operations defined in Section 2.1.1 are generally applicable and hence software-agnostic.", which is confusing to me. The python codes shown in the figures actually use the API operations defined in Sec. 2.1.1. IIUC the executor eventually limits what kind of edit operations are supported by the system, but details are not mentioned. For example, MODIFY() in Fig. 3b seems to support "modification=curve" - what other modifications are supported? In general, I feel that authors could be more upfront about the limitations of the system.
- What are the programmatic limitations of the API? Can it handle multiple bounding boxes (e.g. paint all zebras in the image blue) and/or multiple document elements (e.g. change the national moon day document to a daytime scene)?
- What are the runtime implications of such a complex system design with a feedback mechanism?
- Another limitation compared to approaches operating on the raster image is that the underlying document structure has to be available, so it cannot be applied to any image.

---

> ### Author Rebuttal · Authors · 2024-05-31
>
> We thank Reviewer gr6V for the constructive feedback and acknowledging that our proposed framework is useful for real-world challenges. We address the comments below.
>
>
> __[W0] System breakdown  analysis.__
>
> The suggested aspects are below:
>
> Avg lines of code: 10.75 with 1.4 main API calls.
>
> The syntax error rate is ~33.3% as interpreted by Table 1.
>
> The semantic error rate (number of errors/edits) within our human studies subset is ~34.6% without feedback and ~12.3% with feedback, where the feedback provider reaches 71% accuracy on a held-out test set.
>
>
> __[W0] Limitations and paper clarity.__
>
> We apologize for the potential confusion on the software dependent claims. Our APIs are indeed designed to be generally applicable and software-agnostic. The dependency claim is made to clarify that those APIs are not exactly the final interfaces for manipulating the underneath document structure/contents, where some efforts would be made to map them to the actual execution functions (e.g., mapping from our Python APIs to corresponding  design editing software code.) We will supplement a list of supported actions/edits to each of our API as suggested (e.g., modification supports _curve_, _italic_, _bold_, textual edits, etc.).
>
> __”Grounding”__ means localizing an entity (phrase) in images.
>
>
> __[W1] Programmatic limitations of the system.__
>
> Yes. For the zebras example, the grounding module will propose boxes grounded to the word _“zebra”_, and the edit program will loop over them with applying corresponding actions. Our framework can process multiple contents by compiling a procedure invoking multiple API calls (with proper logic) to the contents.
> However, requests may need to be more specific, e.g., the user needs elaborate the meaning of _”daytime”_ scene (e.g., _”change moon to sun”_) as inferring implicit editing is left as a future exciting work.
>
>
> __[W2] System runtime.__
>
> In practice, the trained feedback provider is launched in a “serving mode” on a separate GPU queuing for inputs, and if all other model weights are in-memory, the overall inference takes roughly 10-15 seconds per each request. The editing software may add another 10-15 seconds for content re-rendering.
>
>
> __[W3] Assumption of document structure.__
>
> We acknowledge such a limitation with an emphasis on presumption of existing editing software here. We foresee a combination of pixel-generative models and our framework as an exciting future direction for achieving even more general use cases.

---

> > ### Comment · Reviewer_gr6V · 2024-06-05
> >
> > Thanks for the response. I'm keeping my positive rating.

---

### Official Review · Reviewer_5Bmb · 2024-05-10

**Rating:** 5
**Confidence:** 3
**Ethics Flag:** 1

**Summary:**

In this paper, a multimodal LLM-agent-based framework, Agent-DocEdit is proposed for the task of automatic (multimodal) document content editing. It consists of four major components:
(1) Edit Program Generator -- takes the user's editing request as input and generates a sequence of editing function calls via LLM code generation.
(2) Multimodal Grounder -- identifies the element IDs in the document that need to be actioned upon, using an architecture derived from phrase grounding models for computer vision tasks.
(3) Executor -- rewrites the edit program to executable programs that modify the document content.
(4) Feedback Provider -- An LLM-based evaluator that compares the executor output against the original request, and generates feedback which is used to instruct the EPG (stage 1) to refine the generated edit program.

**Questions To Authors:**

- How well does the system scale to new tools?
- How well does it scale to new actions (e.g. ROTATE, FLIP, CROP, SCALE, etc.)
- How sensitive is the system to different kinds of prompts?
- There are a lot of incoherent sentences in the paper, e.g. "the Edit Program Generator takes input human user editing request (in
natural language) as input." I'll urge the authors to please review the manuscript to make it more readable.

**Reasons To Accept:**

* This work brings together various components into an end-to-end document editing system, leveraging the tool-augmentation and code generation abilities of LLMs to modify underlying structures of rich documents.
* The system operates directly on the structured representation of a document and produces structured content as output.
* Propose an iterative refinement technique that helps improve the performance. This is a useful way to evaluate and improve the performance.
* The system reports improved performance over the DocEdit baseline in many aspects.

**Reasons To Reject:**

There are a lot of implementation details that are missing, which would make it hard to reproduce the system or reason about the robustness of the results. Here are a few:
* 2.4.1: Dataset construction. How are negative images constructed and verified for quality (i.e. how do you reject bad training examples)? What is the proportion of positive and negative samples? If only minor templates are used, is the model robust to newer kinds of instructions, such as the ones that heavily paraphrase the templates or follow completely different grammatical structures (e.g. "hbd chage color to red", instead of the well-formatted "Change the color of the happy birthday to red")?
* 2.4.1: Feedback-based refinement. How is the quality of the Multimodal Feedback module evaluated?
* 3.1: Error analysis of both action parameters and component parameters is missing. What types of actions does the system do well at, what does it not do well at? What kinds of components are easy for the system to handle?
* 3.3: A lot of the details of human evaluation are missing. Who are the human evaluator? What is the input shown to a human evaluator? What are the guidelines given to the human evaluators? How many annotations are done per test sample? How are conflicts adjudicated?
* Discussion on the limitations of the feedback mechanism seems to be missing. Are there any cases that pass the semantic and syntactic tests yet do not fulfill the request accurately? For example, what if the system tampers with an original text that needs to be modified or changes its font or size in an undesirable way? What about changes to elements not specified in the original request (i.e. maybe some text was asked to be changed but led to a change in other components, etc.)

* It will be useful to see an end-to-end analysis of system robustness and error analysis. Is the system able to recover from noise in the inputs? Which stage of the pipeline is the most robust? Which one is the least robust? If a given input fails at a certain stage, can the later stages ameliorate any issues? A Sankey diagram visualization would be useful.

Overall, while this is an interesting work, but seems to be lacking in terms of novelty. It reuses various pre-existing components from prior works. While that itself is not a problem, the lack of rigor in describing the details or conducting the evaluations limits its potential impact.

---

> ### Author Rebuttal · Authors · 2024-05-31
>
> We thank Reviewer 5Bmb for all suggestions and constructive feedback, where we address them below.
>
>
> __[W1,2] Details of feedback mechanism.__
>
> Negative samples are constructed by manipulating the contents with exactly opposite or wrong actions (we know right/wrong from the ground truths) against a sampled request – guaranteeing the resulting samples contradict it.
> Positive/negative samples have equal amounts.
> We tried paraphrasing the requests with LLMs, however, it did not bring significant benefits.
> The feedback module reaches 71% accuracy on a held-out test set.
>
>
> __[W3,Q1,3] Analysis of actions/components.__
>
> The error rates (w/o feedback) of top 3 frequent actions in the human subset:
>
> REPLACE: 22%
>
> MODIFY: 30%
>
> MOVE: 37%
>
> Our system does particularly well at REPLACE.
> We hypothesize MOVE can benefit from improved target position predictors.
> Our grounder performs quite evenly well on all kinds of components – no particular bias against or towards kinds of components is observed.
>
> For new tools/actions, as long as comprehensive descriptions/examples are provided, our system can generate effective plans. (The grounder and feedback module are agnostic to these.)
> Clearer and more comprehensive guideline prompts give higher quality in the edit programs.
>
>
> __[W4] Details of human studies.__
>
> We recruited 5 college students with linguistic/CS majors and presented them with the edit request alongside outcomes from all 4 methods (scrambled). They ranked them (1-4) on how aligned the samples were to the request. Each instance is ranked by all evaluators, and averaged as in Table 4.
>
>
> __[W5] Limitations of feedback mechanism.__
>
> There are rare cases even semantic/syntactic verifications could overlook, such as complete style changing. Such kinds of requests are not currently dealt with by our system, however, our metrics and human judgements would be able to spot these and score them appropriately.
> The ”wrongly grounded component” in Sec.2.4.1 is exactly designed to handle when the system accidentally changes non-indicated components.
>
>
> __[W6] System robustness.__
>
> The feedback module is proposed exactly for this reason: to make the system capable of recovering from errors. Once the edit program is syntax-bug-free, the semantic errors may come from failed grounding, and the feedback will help search other proposed boxes for more correct components to act on.
> Component robustness can be inferred from Table 1 (code pass rate and parameter accuracies), and Table 2.

---

> > ### Author Response · Authors · 2024-06-06
> >
> > Dear Reviewer 5Bmb,
> >
> > We appreciate the constructive feedback you provided, and have made our best efforts to address all your concerns and suggestions in detail.
> >
> > As the end of the discussion period is approaching, we hope you could kindly let us know if they successfully addressed your concerns and helped you re-evaluate our paper.
> >
> > Please do not hesitate to let us know if you have any further questions! We appreciate your efforts in reviewing our work!
> >
> > Thank you very much!
> >
> > Authors

---

### Official Review · Reviewer_Xzje · 2024-05-11

**Rating:** 7
**Confidence:** 4
**Ethics Flag:** 1

**Summary:**

This paper presents a method to edit visually rich documents by generating code to call APIs. It achieves this by combining multiple components, including SRL, visual grounding, code generation, and execution-based feedback. It improves end-to-end performance of visually rich document editing.

**Reasons To Accept:**

- It improves visual document editing by code generation, which is a novel way to solve this problem
- Modeling complex tasks as agents is an interesting avenue of research that worths more exploration
- Validity of the proposed method and automatic metrics are confirmed with human judgment

**Reasons To Reject:**

- The proposed method works by combining multiple existing components in each stage of processing. More discussions / comparison on the choice of components will be useful.
- Analysis of success and failure examples will be useful too

---

> ### Author Rebuttal · Authors · 2024-05-31
>
> We are thankful to Reviewer Xzje for the comments and that the reviewer thinks our work is novel and the exploration of agents in complex tasks is vital. We address the comments below.
>
> __[W1] Additional discussions on the system design choices.__
>
> Thanks for the suggestion. Below we discuss additional design choices which will be included in the final manuscript.
>
> __Program Generator:__ The edit program generator is built on top of instruction tuned code-gen. models. Our early experiments show that OpenAI’s GPT is better at attending to long and complex API lists (via system prompt), than other concurrent models such as Code-Llama. However, our framework __does not__ depend on specific code-generators and any of them can be used by our framework.
>
> __Grounding Modules:__ In addition to the GLIP used in this work, there are a few other decent grounding models such as Grounding-DINO [1]. We chose GLIP for its simplicity and high adaptability to other data domains. We further improve the phrase extraction with the method in Sec 2.2.
>
> __Feedback Provider:__ The feedback for the editing outcomes requires examination of both the input document contents (images) and the edit requests, and the actual feedback is a generated sentence regarding the success/failure of the editing. Consequently, strong multimodal LLMs, such as LLaVa, naturally emerged as the most suitable candidate for this task.
>
> [1] Liu, Shilong, et al. "Grounding dino: Marrying dino with grounded pre-training for open-set object detection." arXiv 2023.
>
>
> __[W2] Success and failure cases.__
>
> In the final manuscript we will discuss success/failure cases.  Our general observations are: The two (instruction-tuned) generative baselines often exhibit severe degeneration of the parts that are supposed to be intact, or fail to capture the intentions of the user editing request. Our modularized agent framework more successfully interprets the user requests and manipulates the components much more accurately. The feedback mechanism alleviates initial errors from both the edit program generation as well as inaccurate grounding.
>
> We suggest that, for instructed document editing without a ready-to-use editing software (as in our work), an interesting future work could be an integration of our modularized framework into the instruction-tuned generative pipeline, making it more robust and faithful to the input requests.

---

> > ### Comment · Reviewer_Xzje · 2024-06-04
> >
> > Thanks for the informative feedback! I keep the positive review of the paper based on the feedback.

---

### Decision · Program_Chairs · 2024-07-10

**Decision:**

Accept

**Comment:**

This paper presents a method to edit visually rich documents by generating code to call APIs. After author rebuttal, it received scores of 5679. On one hand, reviewers agree that (1) the proposed method to consider this as a tool-use agent problem is interesting, convincing and novel, compared with end-to-end diffusion models; and (2) experiments show the effectiveness of the method, both in terms of automatic metrics and human evaluation.

On the other hand, the drawbacks of the paper include that (1) many implementation details are missing, and the proposed method can be hard to reproduce; (2) more ablations are needed. There are many moving parts to the system, and breaking it down would give valuable insights regarding all the design decisions that eventually led to the proposed architecture.

The authors have done a relatively good job of rebuttal. Given the overall positive score with one strong support from one reviewer, the AC thinks that the merits of the paper outweigh the flaws, and decided to recommend acceptance by the end.